TECHNICAL RELEASE

# CompactTree: a lightweight header-only C++ library and Python wrapper for ultra-large phylogenetics

Niema Moshiri[1,*]

1 Department of Computer Science & Engineering, UC San Diego, La Jolla, CA 92093, USA

## ABSTRACT

The study of viral and bacterial species requires the ability to load and traverse ultra-large phylogenies with tens of millions of tips, but existing tree libraries struggle to scale to these sizes. We introduce CompactTree, a lightweight header-only C++ library with a user-friendly Python wrapper for traversing ultra-large trees that can be easily incorporated into other tools. We show that CompactTree is orders of magnitude faster and requires orders of magnitude less memory than existing tree packages. CompactTree is freely accessible as an open source project: https://github.com/niemasd/CompactTree

**Subjects** Software and Workflows, Bioinformatics, Virology

**Submitted:** 14 December 2024

\* E-mail: niema@ucsd.edu

Preprint submitted at https://doi.org/10.1101/2024.07.15.603593

## STATEMENT OF NEED

As sequencing technologies continue to improve and sequencing costs continue to fall, the amount of viral and bacterial sequencing data available to biologists and epidemiologists continues to rapidly grow. For example, the Global Initiative on Sharing All Influenza Data (GISAID) database currently contains over 16 million SARS-CoV-2 genome sequences collected from all over the world [1]. The explosion of available sequencing data led to a push for more scalable phylogenetic inference tools [2, 3], which has resulted in the creation of ultra-large viral and bacterial phylogenies with tens of millions of tips [4].

Given a viral or bacterial phylogeny, many biological and epidemiological analyses can be reduced to performing one or more traversals along the tree, such as clustering [5], ancestral state reconstruction and molecular clock estimation [6], and transmission risk prediction [7]. Rather than re-implementing standard tree operations from scratch, many phylogenetic tools utilize existing packages that can load and traverse trees, and the tools implement their algorithms using functions provided by these packages. To our knowledge, the following packages are most typically used: Bio++ (C++) [8], Genesis (C++) [9], Biopython (Python) [10], DendroPy (Python) [11], ETE 4 (Python) [12], TreeSwift (Python) [13], bigtree (Python) [14], bp (Python) [15], scikit-bio (Python) [16], and ape (R, RRID:SCR_017343) [17]. Of note, most of the aforementioned packages have multiple dependencies, meaning users of tools that utilize these packages will need to have the ability to install all of the dependencies as well.

While these packages work well for analyzing phylogenies with tens or even hundreds of thousands of tips, they face prohibitively large memory usage and runtime when analyzing phylogenies with tens of millions of tips, such as the Greengenes2 tree [4]. To address these

scalability issues while maintaining ease of incorporation into external code bases, we introduce CompactTree, a lightweight header-only C++ library with a user-friendly Python wrapper for traversing ultra-large trees, and we compare its performance against that of existing tree packages.

## IMPLEMENTATION

CompactTree is a header-only C++ library that has no dependencies beyond the C++ Standard Template Library, meaning it can simply be dropped into any C++ code base and be utilized with a single `#include` statement. This is critical because, to our knowledge, all existing C++ libraries for loading and traversing phylogenetic trees, as well as many existing libraries in other programming languages (e.g., Python and R), typically require many other dependencies, which can be prohibitively difficult for users to install. We also provide a user-friendly Python wrapper, the `CompactTree` package on PyPI (RRID:SCR_023145), which can easily be installed via `pip install CompactTree` and utilized in any Python codebase. Documentation of the `compact_tree` class provided by CompactTree is automatically generated upon every update to the CompactTree GitHub repository using Doxygen and can be found online: https://niema.net/CompactTree.

CompactTree is currently only able to load trees from Newick files. Currently, CompactTree supports the following functionalities: iterators to perform standard tree traversals (pre-order, in-order, post-order, and level-order), an iterator over the leaves of the tree, an iterator over the children of a given node, calculating basic properties of a tree (e.g., number of nodes, number of leaves, number of internal nodes, total branch length, average branch length), calculating pairwise distances between nodes, extracting subtrees, finding the Most Recent Common Ancestor of a collection of nodes, and manipulating node labels and branch lengths.

CompactTree internally represents nodes as unsigned integers (`CT_NODE_T`) between 0 and $N - 1$, where $N$ is the number of nodes in the tree. By default, CompactTree uses 32-bit unsigned integers (i.e., `std::uint32_t`), which can safely support trees with up to $N = 2^{32} - 1$ nodes (0 through $2^{32} - 2$; the node value $2^{32} - 1$ is reserved to denote a "null" node), but this can be overridden to use 64-bit unsigned integers (i.e., `std::uint64_t`) by adding the compilation flag `-DCT_NODE_64`. CompactTree loads phylogenetic trees so that every node has a smaller numerical value than all of its children, meaning pre-order and post-order traversals can be trivially defined as iterating over the integer ranges $[0, N - 1]$ and $[N - 1, 0]$, respectively, which is extremely fast (Figure 1c and d).

CompactTree internally represents edge lengths as floating point numbers (`CT_LENGTH_T`). By default, CompactTree uses the `float` type, which generally has 7 decimal digits of precision. However, this can be overridden to use the `double` type, which generally has 15 decimal digits of precision, by adding the compilation flag `-DCT_LENGTH_DOUBLE`.

CompactTree internally represents the tree structure itself using 4 member variables: `parent`, `children`, `label`, and `length`:

(i)   The `parent` variable has type `std::vector<CT_NODE_T>`, and it represents parent relationships: `parent[u]` denotes the parent of node `u`.

(ii)  The `children` variable has type `std::vector<std::vector<CT_NODE_T>>`, and it represents the child relationships: `children[u]` contains all of the children of node `u`.

(iii) The `label` variable has type `std::vector<std::string>`, and it represents node labels: `label[u]` denotes the label of node `u`.



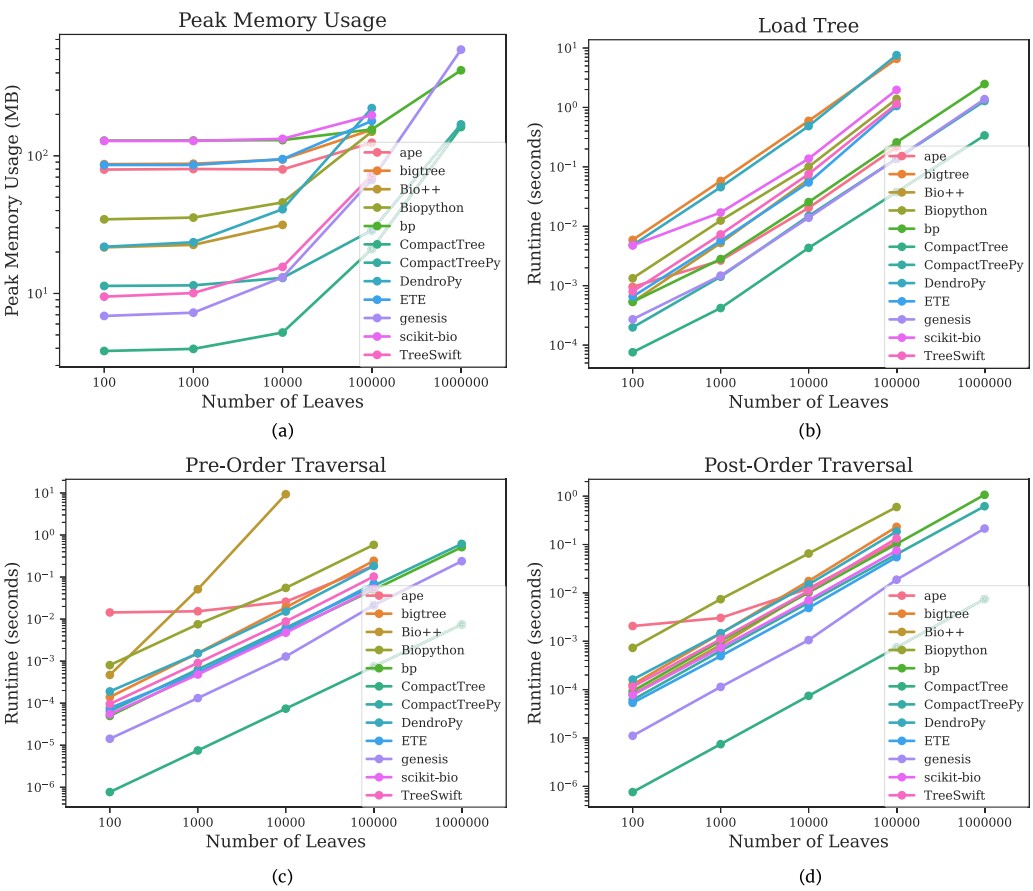

**Figure 1.** Benchmark results. Measurements of (a) peak memory usage as well as runtime of (b) loading a tree, (c) pre-order traversal, and (d) post-order traversal. Each point is the average of 10 runs, and bars depicting the 95% confidence interval are plotted but are too small to see beyond the average points.

(iv) The `length` variable has type `std::vector<CT_LENGTH_T>`, and it represents edge lengths: `label[u]` denotes the length of the edge incident to node `u` (i.e., the length of the edge from `parent[u]` to `u`).

This tree representation scheme is extremely memory-efficient: trees represented using CompactTree are typically orders of magnitude more memory-efficient than trees represented using existing tree packages (Figure 1a). CompactTree is able to provide even further memory savings by allowing users to omit node labels (e.g., for applications that calculate metrics across the entire tree) and/or edge lengths (e.g., for applications that only require the tree topology) when loading a tree. Furthermore, by representing the tree topology using array-based structures in which elements are stored contiguously in memory, CompactTree benefits from cache-friendliness due to memory locality.

To compare the performance of CompactTree against existing tree packages, a benchmarking experiment was performed on binary trees with *n* = 100, 1K, 10K, 100K, and 1M leaves randomly generated under the Yule model [18] using the Dual-Birth Simulator [19]. Trees were loaded into memory, pre-order and post-order traversals were executed, and peak memory usage was measured. The following tools were included in the

benchmark: CompactTree v0.0.8, Bio++ bpp-phyl commit 55e5b45 [8], Genesis v0.32.0 [9], Biopython v1.83 [10], DendroPy v5.0.1 [11], ETE 4 commit a88743b [12], TreeSwift v1.1.45 [13], bigtree v0.24.0 [14], bp v1.0.7 [15], scikit-bio v0.6.2 [16], and ape v5.8-1 [17]. To ensure reproducibility, the benchmark was performed single-threaded via automated GitHub Actions:

- **Repository:** https://github.com/niemasd/CompactTree-Paper
- **Operating System:** Ubuntu 22.04.4 LTS
- **CPU:** AMD EPYC 7763 at 3.2 GHz
- **RAM:** 16 GB.

## RESULTS AND DISCUSSION

As can be seen in Figure 1, CompactTree is orders of magnitude faster and has orders of magnitude lower peak memory usage than existing tree packages. Furthermore, CompactTree was able to load the entire Greengenes2 [4] 2022.10 release ($N$ = 22,090,656 nodes; $n$ = 21,074,441 leaves) in just 2.82 seconds with a peak memory usage of just 1.7 GB, and it was traversed in pre-order and post-order in just 0.044 seconds each.

Notably, ultra-large trees such as the Greengenes2 tree are too large to be loaded into memory by existing tree packages on typical consumer computer hardware, whereas CompactTree is able to load this tree within the memory constraints of even relatively low-cost devices (e.g., Raspberry Pi). Thus, the reimplementation of existing tools, such as TreeCluster [5], to utilize CompactTree would enable phylogenetic analyses, such as molecular clustering on ultra-large trees like the Greengenes2 tree.

However, in its current state, it is less flexible than existing tree packages: CompactTree only supports the Newick file format, whereas other tools support more complex formats such as Nexus and NeXML. Also, while CompactTree allows users to edit node labels and edge lengths, it does not allow modifications to the tree topology in order to ensure the pre-order ordering of the nodes of the tree in their unsigned integer representations, which is critical to CompactTree's speed and memory efficiency.

As mentioned previously, in its current state, CompactTree represents child relationships via `std::vector<std::vector<CT_NODE_T>>`: In other words, every node has its own `std::vector` to store all of its children. Each individual `std::vector` has some amount of memory overhead, which can potentially be optimized in the future. One possible approach is to have a single `std::vector` storing the children of all nodes, such that the children of any given node are stored contiguously, with a secondary data structure (e.g., `std::vector<std::pair>`) to denote the start and end indices of each node's children.

An even more memory-optimized approach to representing child relationships would be enabled by ordering the integer values of the nodes of the tree in a level-order ordering (i.e., 0 is the root, followed by the children of the root, followed by the children of the first child of the root, etc.) rather than the current Depth-First Search (DFS) pre-order ordering that is obtained by parsing the Newick string from left to right. Specifically, in the current representation, while it is guaranteed that any given node will have a smaller integer value than all of its children, there are no guarantees regarding where its children will appear, and the children of a given node will actually likely not be represented as contiguous integers. However, if the integer values of the nodes were represented via level-order ordering, all of the children of any given node would have contiguous integer values,



meaning all of the children could be represented by just storing 2 integers: the minimum and maximum child values. However, to our knowledge, determining a level-order ordering of the nodes in a tree would require either loading the entire Newick string into memory up-front (which would yield significant memory overhead in the constructor) or performing multiple passes over the tree file on disk (which would yield significant slowdown due to additional disk accesses). On the other hand, the current DFS pre-order approach to loading a tree from a file is able to do so via a single pass through the file while only storing a small constant-size buffer of the Newick string in memory at any given time, which is incredibly fast and extremely small memory overhead beyond the tree object itself. In future works, we will continue to explore this line of thinking, and we will explore possible low-memory intermediate representations of the tree topology, such as that proposed by Cordova and Navarro [20], upon which we could perform level-order traversal in order to reorder the nodes of the CompactTree representation.

In summary, we introduce CompactTree, a lightweight header-only C++ library with a user-friendly Python wrapper for traversing ultra-large trees that can be easily incorporated into other tools. CompactTree is orders of magnitude faster and requires orders of magnitude less memory than existing tree packages, and we hope CompactTree will aid bioinformaticians in their efforts to build massively-scalable tools for analyzing ultra-large phylogenetic trees.

## AVAILABILITY OF SOURCE CODE AND REQUIREMENTS

- Project name: CompactTree
- Project home page: https://github.com/niemasd/CompactTree
- Operating system(s): Platform independent
- Programming language: C++ and Python
- Other requirements: None
- License: GNU GPL 3.0
- RRID:SCR_026497.

## DATA AVAILABILITY

Snapshots of the CompactTree software [21] and code to generate the paper data and figures [22] are archived in SoftWare Heritage. Additional supporting data is available in GigaDB [23].

## ABBREVIATIONS

DFS, Depth-First Search; GISAID, Global Initiative on Sharing All Influenza Data.

## DECLARATIONS

### Ethical approval

The authors declare that ethical approval was not required for this type of research.

### Competing interests

The authors declare that they have no competing interests.

## Author's contributions

NM: Conceptualization, Data curation, Formal analysis, Funding acquisition, Investigation, Methodology, Project administration, Software, Resources, Supervision, Validation, Visualization, Writing (original draft), Writing (review & editing).

## Funding

This work was supported by UC San Diego Faculty Research Funds.

## Acknowledgements

We would like to thank Siavash Mirarab and Ali Osman Berk Sapci for fruitful conversations, particularly in the context of potential future improvements to reduce memory usage via alternative representations of child relationships.

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
