## [Editor Report]

Editor’s AssessmentAs volumes of viral and bacterial sequence data grow exponentially, the field of computational phylogenetics now demands resources to manage the burgeoning scale of this input data. This study introduces CompactTree, a C++ library designed for ultra-large phylogenetic trees with millions of tips. To address these scalability issues while maintaining ease of incorporation into external code bases, CompactTree is a header-only library with enhanced performance utilizing minimal dependencies, optimized node representation, and memory-efficient tree structure schemes. Resulting in significantly reduced memory footprints and improved processing times. Peer review requested some more detail on the functionality and some real-world examples, demonstrating the current utility of the tool. Although primarily supporting the (text-based) Newick format, the increased and extensibility scalability holds promise for multiple biological and epidemiological applications supporting more complex formats such as Nexus and NeXML. The tool is open source (GPLv3 licensed) and available in GitHub: https://niema.net/CompactTreeEditor’s AssessmentAs volumes of viral and bacterial sequence data grow exponentially, the field of computational phylogenetics now demands resources to manage the burgeoning scale of this input data. This study introduces CompactTree, a C++ library designed for ultra-large phylogenetic trees with millions of tips. To address these scalability issues while maintaining ease of incorporation into external code bases, CompactTree is a header-only library with enhanced performance utilizing minimal dependencies, optimized node representation, and memory-efficient tree structure schemes. Resulting in significantly reduced memory footprints and improved processing times. Peer review requested some more detail on the functionality and some real-world examples, demonstrating the current utility of the tool. Although primarily supporting the (text-based) Newick format, the increased and extensibility scalability holds promise for multiple biological and epidemiological applications supporting more complex formats such as Nexus and NeXML. The tool is open source (GPLv3 licensed) and available in GitHub: https://niema.net/CompactTree

---

## [Reviewer Report]

Reviewer name and names of any other individual's who aided in reviewerJeet SukumaranDo you understand and agree to our policy of having open and named reviews, and having your review included with the published manuscript. (If no, please inform the editor that you cannot review this manuscript.)YesIs the language of sufficient quality?YesPlease add additional comments on language quality to clarify if neededIs there a clear statement of need explaining what problems the software is designed to solve and who the target audience is? YesAdditional CommentsIs the source code available, and has an appropriate Open Source Initiative license <a href="https://opensource.org/licenses" target="_blank">(https://opensource.org/licenses)</a> been assigned to the code?YesAdditional CommentsAs Open Source Software are there guidelines on how to contribute, report issues or seek support on the code?YesAdditional CommentsIs the code executable?YesAdditional CommentsIs installation/deployment sufficiently outlined in the paper and documentation, and does it proceed as outlined?YesAdditional CommentsIs the documentation provided clear and user friendly?YesAdditional CommentsExcellent documentation. A pleasure to read.Is there enough clear information in the documentation to install, run and test this tool, including information on where to seek help if required?YesAdditional CommentsIs there a clearly-stated list of dependencies, and is the core functionality of the software documented to a satisfactory level?YesAdditional CommentsHave any claims of performance been sufficiently tested and compared to other commonly-used packages? YesAdditional CommentsIs test data available, either included with the submission or openly available via cited third party sources (e.g. accession numbers, data DOIs)?Additional CommentsDidn't checkAre there (ideally real world) examples demonstrating use of the software? NoAdditional CommentsIs automated testing used or are there manual steps described so that the functionality of the software can be verified?Additional CommentsAny Additional Overall Comments to the AuthorRecommendationAccept

---

## [Reviewer Report]

Reviewer name and names of any other individual's who aided in reviewerZiqi DengDo you understand and agree to our policy of having open and named reviews, and having your review included with the published manuscript. (If no, please inform the editor that you cannot review this manuscript.)YesIs the language of sufficient quality?YesPlease add additional comments on language quality to clarify if neededIs there a clear statement of need explaining what problems the software is designed to solve and who the target audience is? YesAdditional CommentsIs the source code available, and has an appropriate Open Source Initiative license <a href="https://opensource.org/licenses" target="_blank">(https://opensource.org/licenses)</a> been assigned to the code?YesAdditional CommentsAs Open Source Software are there guidelines on how to contribute, report issues or seek support on the code?YesAdditional CommentsIs the code executable?YesAdditional CommentsIs installation/deployment sufficiently outlined in the paper and documentation, and does it proceed as outlined?YesAdditional CommentsI'm able to run all the tests and used CompactTree c++ correctly except for encounter issue installation installing Python Wrapper via pip install CompactTree.Is the documentation provided clear and user friendly?YesAdditional CommentsIs there enough clear information in the documentation to install, run and test this tool, including information on where to seek help if required?NoAdditional CommentsI'm able to run all the tests and used CompactTree c++ correctly except for the Python Wrapper. I encountered via pip install CompactTree. I would recommend providing clearer instructions on installing the Python wrapper, ideally also the environment for testing.Is there a clearly-stated list of dependencies, and is the core functionality of the software documented to a satisfactory level?YesAdditional CommentsHave any claims of performance been sufficiently tested and compared to other commonly-used packages? YesAdditional CommentsIs test data available, either included with the submission or openly available via cited third party sources (e.g. accession numbers, data DOIs)?YesAdditional CommentsAre there (ideally real world) examples demonstrating use of the software? YesAdditional CommentsCompactTree has provided examples of simulated trees for testing comparing to other peer packages. In the meanwhile it mentioned its ability to load the ~22M nodes greengenes2 tree. It would be great to see the test workflow so users can verify.Is automated testing used or are there manual steps described so that the functionality of the software can be verified?YesAdditional CommentsAny Additional Overall Comments to the AuthorCompactTree is aimed at a very specific task, that of loading large phylogenetic trees with millions of nodes. The result shows that it is significantly faster than the other peer tools not only in loading but also in traversing trees, with less peak memory usage. It also includes the test workflow for users to repeat the test in comparison with other peer tools.RecommendationMinor Revisions

---

## [Reviewer Report]

Reviewer name and names of any other individual's who aided in reviewerGiorgio BianchiniDo you understand and agree to our policy of having open and named reviews, and having your review included with the published manuscript. (If no, please inform the editor that you cannot review this manuscript.)YesIs the language of sufficient quality?YesPlease add additional comments on language quality to clarify if neededIt is slightly confusing that the paper is written using plural pronouns ("We"), when there is a single author.Is there a clear statement of need explaining what problems the software is designed to solve and who the target audience is? NoAdditional CommentsThe statement of need is present; however, it does not clearly explain what kinds of problems the software will be able to solve, beyond generic statements about addressing scalability issues. The aims of the library should be explored in more detail: as noted by the author, this library offers great speed and efficiency, but at the cost of reduced flexibility and functionality compared to other tools. Speed and efficiency are always good things, but what does the library actually do? A very fast library that does nothing is not particularly useful. So, what specific analyses does CompactTree allow, that would be impractical using other tools? For example, they could select a case study from the literature, where the analyses were limited by the algorithm, and use their library to extend the analysis to a larger dataset. The author mentions clustering, ancestral state reconstruction, and transmission risk prediction as examples of analyses that involve tree traversals, so they could start here (although I am not convinced that the efficiency of the tree representation is the computational bottleneck in these cases). The results should also be briefly mentioned in the abstract. Furthermore, the author mentions a number of packages used to analyse trees, but these are all Python packages. Since CompactTree is presented as a C++ library, this seems odd; other tools and programming languages should be mentioned/compared. For example, “ape” and “phytools” are very popular R packages, while “Bio++” is another C++ library; a literature review (or a simple web search) may reveal other such libraries. Also, the reference given for bp (“[4]”) is incorrect.Is the source code available, and has an appropriate Open Source Initiative license <a href="https://opensource.org/licenses" target="_blank">(https://opensource.org/licenses)</a> been assigned to the code?YesAdditional CommentsAs Open Source Software are there guidelines on how to contribute, report issues or seek support on the code?YesAdditional CommentsIs the code executable?YesAdditional CommentsIs installation/deployment sufficiently outlined in the paper and documentation, and does it proceed as outlined?YesAdditional CommentsEverything works fine if the header is included in a single source file, but if multiple distinct files contain the #include statement, a compilation error will occur due to the multiple definitions. In a real-world application, the library would reasonably need to be included in multiple source files, so this should be fixed.Is the documentation provided clear and user friendly?YesAdditional CommentsThe documentation "Cookbook" is very nicely organised.Is there enough clear information in the documentation to install, run and test this tool, including information on where to seek help if required?YesAdditional CommentsIs there a clearly-stated list of dependencies, and is the core functionality of the software documented to a satisfactory level?YesAdditional CommentsHave any claims of performance been sufficiently tested and compared to other commonly-used packages? NoAdditional CommentsWhile the author compares CompactTree to a number of Python packages, no comparison is made against tools that use other programming languages. In particular, the author states that there is no C++ library for loading and traversing phylogenetic trees; however, as I mentioned, at least Bio++ exists and appears to be reasonably well cited. Furthermore, the memory plot does not consider the baseline memory usage. This is evident in the first two datapoints (n=100 and n=1000) for each tool, which show a very small difference, despite the leaf count increasing by an order of magnitude. If the first datapoint is subtracted from all subsequent datapoints, the memory plot looks quite similar to the other plots. If you re-run the benchmarks to include other tools, I would suggest including a “control” datapoint with a very small n (or even, loading the library without opening a tree), and subtracting this from all other datapoints; this will provide an estimate of the memory actually used to load the trees.Is test data available, either included with the submission or openly available via cited third party sources (e.g. accession numbers, data DOIs)?YesAdditional CommentsAre there (ideally real world) examples demonstrating use of the software? NoAdditional CommentsAs I mentioned above, having at least one example demonstrating an analysis that is significantly improved by the use of this library would be beneficial. Discussion of the improvements should also consider usability trade-offs in a real-world scenario.Is automated testing used or are there manual steps described so that the functionality of the software can be verified?NoAdditional CommentsAny Additional Overall Comments to the AuthorThe library looks promising and is reasonably well documented, the only two things that are really missing are a real-world practical application and a comparison with other relevant alternatives (especially Bio++). A large portion of the manuscript is spent describing how the library could be improved, rather than what it can currently do. This could be summarised in just one or two sentences, thus leaving more space for describing the real-world example.RecommendationMajor Revisions